# VARIATIONAL MULTI-TASK LEARNING

## ABSTRACT

Multi-task learning aims to improve the overall performance of a set of tasks by leveraging their relatedness. When training data is limited using priors is pivotal, but currently this is done in ad-hoc ways. In this paper, we develop *variational multi-task learning* - VMTL, a general probabilistic inference framework for simultaneously learning multiple related tasks. We cast multi-task learning as a variational Bayesian inference problem, which enables task relatedness to be explored in a principled way by specifying priors. We introduce Gumbel-softmax priors to condition the prior of each task on related tasks. Each prior is represented as a mixture of variational posteriors of other related tasks and the mixing weights are learned in a data-driven manner for each individual task. The posteriors over representations and classifiers are inferred jointly for all tasks and individual tasks are able to improve their performance by using the shared inductive bias. Experimental results demonstrate that VMTL is able to tackle challenging multi-task learning with limited training data well, and it achieves state-of-the-art performance on four benchmark datasets consistently surpassing previous methods.

## 1 INTRODUCTION

Multi-task learning (Caruana, 1997) is a fundamental learning paradigm for machine learning, which aims to simultaneously solve multiple related tasks to improve the performance of the individual tasks by sharing knowledge. The crux of multi-task learning is how to explore task relatedness (Argyriou et al., 2007; Zhang & Yeung, 2012), which is non-trivial since the underlying relationship among tasks can be complicated and highly nonlinear. This has been extensively investigated in previous work by learning shared features, designing regularizers imposed on parameters (Pong et al., 2010; Kang et al., 2011; Jawanpuria et al., 2015; Jalali et al., 2010) or exploring priors over parameters (Heskes, 2000; Bakker & Heskes, 2003; Xue et al., 2007; Zhang & Yeung, 2012; Long et al., 2017). Recently, deep neural networks have been developed learning shared representations in the feature layers while keeping the classifier layers independent (Yang & Hospedales, 2016; Hashimoto et al., 2016; Ruder et al., 2017). It would be beneficial to learn them jointly by fully leveraging the shared knowledge related tasks, which however remains an open problem.

In our work, we consider a particularly challenging setting, where each task contains limited training data. Even more challenging, we have only a handful of related tasks to gain shared knowledge from. This is in stark contrast to few-shot learning (Gordon et al., 2018; Finn et al., 2017; Vinyals et al., 2016) that also suffers from limited data for each task, but usually have a large number of related tasks. Therefore, in our scenario, it is difficult to learn a proper model for each task independently without overfitting (Long et al., 2017; Zhang & Yang, 2017) and it is crucial to leverage the inductive bias (Baxter, 2000) provided by various other related tasks that are learned simultaneously. To do so, we employ the Bayesian framework as it is able to deliver uncertainty estimates on predictions and automatic model regularization (MacKay, 1992; Graves, 2011), which makes it well suited for multi-task learning with limited training data. The major motivation of our work is to leverage the Bayesian learning framework to handle the great challenges of limited data in multi-task learning.

In this paper, we introduce *variational multi-task learning* - VMTL, a novel variational Bayesian inference approach that can explore task relatedness in a principled way. In order to fully utilize the shared knowledge from related tasks, we explore task relationships in both the feature representation and the classifier by placing prior distributions over them in a Bayesian framework. Thus, multi-task learning is cast as a variational inference problem for feature representations and classifiers jointly. The introduced variational inference allows us to specify the priors by depending on variational pos-

teriors of related tasks. To further leverage the shared knowledge from related tasks, we introduce the Gumbel-softmax prior to each task, which is a mixture of variational posteriors of other related tasks. We apply the optimization technique (Jang et al., 2016) to learn the mixing weights jointly with the probabilistic modelling parameters by back-propagation. The Gumbel-softmax priors are incorporated into the inference of posteriors over representations and classifiers, which enable them to leverage the shared knowledge. We validate the effectiveness of the proposed VMTL by extensive evaluation on four challenging benchmarks for multi-task learning. The results demonstrate the benefit of variational Bayesian inference for modeling multi-task learning. VMTL consistently outperforms previous methods in terms of the average accuracy of all tasks.

## 2 METHODOLOGY

In this work, we tackle the challenging multi-task learning setting where only a few training samples are available for each task, and only a limited number of related tasks to share knowledge. We investigate multi-task learning under the Bayesian learning framework, where we learn the task relationship in a principled way by exploring priors. We cast multi-task learning as a variational inference problem, which offers a unified framework to learn task relatedness in both feature representations and task-specific classifiers.

### 2.1 MULTI-TASK VARIATIONAL INFERENCE

In our setting the tasks are classification problems which share the same label space, but where the samples are drawn from different data distributions. Given a set of related tasks $\{\mathcal{D}_t\}_{t=1}^T$ and each task $\mathcal{D}_t = \{\mathbf{x}_t^n, \mathbf{y}_t^n\}_{n=1}^{N_t}$, $N_t$ is the number of training samples in the $t$-th task, the goal under this setting is to predict the label $\mathbf{y}$ of the test sample $\mathbf{x}$ for all tasks simultaneously, using the shared information extracted from other related tasks. We note that the main challenge is the limited number of labeled samples for each task, which makes it difficult to learn a proper model for each task independently (Long et al., 2017; Zhang & Yang, 2017).

Under this multi-task learning setting, we consider the Bayesian treatment. For a single task without knowledge sharing from related tasks, we place a prior over its classifier parameter $\mathbf{w}$, which gives rise to the following data log-likelihood to maximize:

$$\log p(\mathcal{D}) = \log \int p(\mathcal{D}|\mathbf{w}) p(\mathbf{w}) d\mathbf{w}. \tag{1}$$

For multi-task learning, we solve $T$ tasks simultaneously with knowledge sharing among tasks. Thus, after observing data from all $T$ tasks, the true posterior $p(\mathbf{w}_t|\mathcal{D}_t)$ of a single task $t$ becomes $p(\mathbf{w}_t|\mathcal{D}_{1:T})$. Using Bayes' rule, we have the posterior for task $t$ as follows:

$$p(\mathbf{w}_t|\mathcal{D}_{1:T}) \propto p(\mathbf{w}_t) \prod_{i=1}^T p(\mathcal{D}_i|\mathbf{w}_i) \propto p(\mathbf{w}_t|\mathcal{D}_{1:T} \backslash \mathcal{D}_t) p(\mathcal{D}_t|\mathbf{w}_t). \tag{2}$$

We introduce a variational distribution $q(\mathbf{w}_t; \theta_t)$ parameterized by $\theta_t$ for current task $t$ to approximate the true posterior, which involves minimizing the Kullback-Leibler (KL) divergence between the variational distribution and the true posterior:

$$\theta_t^* = \arg\min_{\theta_t} \mathbb{D}_{\mathrm{KL}}\Big[ q(\mathbf{w}_t; \theta_t) || p(\mathbf{w}_t|\mathcal{D}_{1:T}) \Big]. \tag{3}$$

Generally, the approximate posterior is defined as a fully-factorized Gaussian distribution, i.e. $q(\mathbf{w}_t; \theta_t) = \mathcal{N}(\boldsymbol{\mu}_t, \boldsymbol{\sigma}_t^2)$ (Nguyen et al., 2017; Kingma et al., 2015; Molchanov et al., 2017). By applying Eq. (2) into (3) and extending them to all tasks, we obtain an evidence lower bound (ELBO) for multi-task learning:

$$\mathcal{L}_{\mathrm{ELBO}}(\boldsymbol{\theta}) = \frac{1}{T} \sum_{t=1}^T \left[ \mathbb{E}_{\mathbf{w}_t \sim q}\Big[ \log p(\mathcal{D}_t|\mathbf{w}_t) \Big] - \mathbb{D}_{\mathrm{KL}}\Big[ q(\mathbf{w}_t; \theta_t) || p(\mathbf{w}_t|\mathcal{D}_{1:T} \backslash \mathcal{D}_t) \Big] \right], \tag{4}$$

where $\boldsymbol{\theta} = \{\theta_t\}_{t=1}^T$ is the set of parameters for all task-specific classifiers. We maximize the ELBO to optimize the model parameters for multi-task learning. It is worth noting that for MTL the prior of

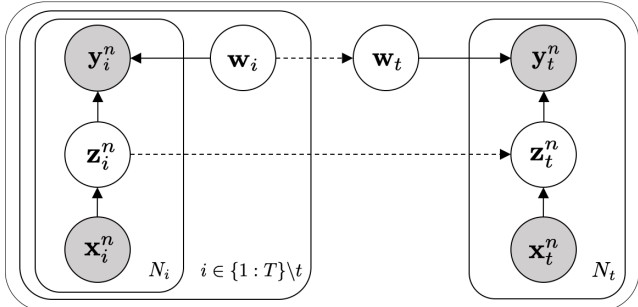

Fig. 1. A graphical illustration of the proposed model, variational multi-task learning (VMTL). The two dashed lines show the prior of current task depends on the posteriors of other tasks for classifiers and representations. VMTL offers a principled way to explore task relationships: for each task, the priors over the classifiers and feature representations are conditioned on other tasks.

each task is conditioned on other tasks which allows knowledge sharing between them, in contrast to single-task learning without knowledge sharing where uninformative Gaussian priors are generally applied. Actually, this multi-task ELBO in Eq. (4) provides a general probabilistic inference framework that enables task relatedness to be explored in a principled way by leveraging the inductive bias provided by other related tasks (Ruder, 2017). For each task, the conditional prior $p(\mathbf{w}_t|\mathcal{D}_{1:T}\backslash\mathcal{D}_t)$ will serve as a regularizer for the posterior inference.

In order to fully leverage the shared knowledge from related tasks to improve each individual task, in addition to the classifiers, we would also like to share knowledge among the feature representations of samples from different tasks. To this end, we introduce the conditional prior $p(\mathbf{z}_t^n|\mathbf{x}_t^n)$ over the feature representation $\mathbf{z}_t^n$ for each sample $\mathbf{x}_t^n$ in task $t$. In doing so, we are able to explore task relatedness among feature representations in a unified way, as we do for classifiers.

To this end, we rewrite the log-likelihood as a sum over the marginal likelihoods of individual data points as follows:

$$\log p(\mathcal{D}_t|\mathbf{w}_t) = \frac{1}{N_t}\sum_{n=1}^{N_t}\log p(\mathbf{y}_t^n|\mathbf{w}_t,\mathbf{x}_t^n). \tag{5}$$

The variational Bayesian inference framework developed in Eq. (4) allows the latent representation to be seamlessly incorporated into the log-likelihood in Eq. (5). Under the assumption that $\mathbf{w}_t$ and $\mathbf{z}_t^n$ are conditionally independent, we therefore obtain a new marginal conditional log-likelihood as follows:

$$\log p(\mathbf{y}_t^n|\mathbf{w}_t,\mathbf{x}_t^n) = \log\int p(\mathbf{y}_t^n,\mathbf{z}_t^n|\mathbf{w}_t,\mathbf{x}_t^n)d\mathbf{z}_t^n = \log\int p(\mathbf{y}_t^n|\mathbf{z}_t^n,\mathbf{w}_t)p(\mathbf{z}_t^n|\mathbf{x}_t^n)d\mathbf{z}_t^n. \tag{6}$$

The posterior $p(\mathbf{z}_t^n|\mathbf{x}_t^n,\mathbf{y}_t^n)$ of latent representations $\mathbf{z}_t$ is intractable. Therefore, like VAEs (Kingma & Welling, 2013) we introduce again a variational posterior namely $q(\mathbf{z}_t|\mathbf{x}_t;\phi)$, which is made conditional on the sample $\mathbf{x}_t^n$. We note that $\phi$ is the parameters of the inference network for latent representations, which is optimized jointly with other parameters in our model. By introducing $q(\mathbf{z}_t^n|\mathbf{x}_t^n;\phi)$ into Eq. (6) and applying Jensen's inequality, we have

$$\log p(\mathbf{y}_t^n|\mathbf{w}_t,\mathbf{x}_t^n) \geq \mathbb{E}_{\mathbf{z}_t^n\sim q_{\mathbf{z}_t^n}}\Big[\log p(\mathbf{y}_t^n|\mathbf{z}_t^n,\mathbf{w}_t)\Big] - \mathbb{D}_{\mathrm{KL}}\Big[q(\mathbf{z}_t^n|\mathbf{x}_t^n;\phi)||p(\mathbf{z}_t^n|\mathbf{x}_t^n)\Big]. \tag{7}$$

After integrating Eq. (4) and (7), we obtain the following variational objective for multi-task learning:

$$\mathcal{L}_{\mathrm{VMTL}}(\boldsymbol{\theta},\phi) = \frac{1}{T}\sum_{t=1}^{T}\Bigg[\frac{1}{N_t}\sum_{n=1}^{N_t}\Big[\mathbb{E}_{q_{\mathbf{w}_t}}\mathbb{E}_{q_{\mathbf{z}_t^n}}\big[\log p(\mathbf{y}_t^n|\mathbf{z}_t^n,\mathbf{w}_t)\big] - \mathbb{D}_{\mathrm{KL}}\big[q(\mathbf{z}_t^n|\mathbf{x}_t^n;\phi)||p(\mathbf{z}_t^n|\mathbf{x}_t^n)\big]\Big]$$
$$- \mathbb{D}_{\mathrm{KL}}\big[q(\mathbf{w}_t;\theta_t)||p(\mathbf{w}_t|\mathcal{D}_{1:T}\backslash\mathcal{D}_t)\big]\Bigg]. \tag{8}$$

The objective function provides a general probabilistic inference framework for multi-task learning, which allows us to jointly explore shared knowledge among representations and classifiers in a

unified way by specifying priors. The detailed derivation of the ELBO for $\mathbf{z}$ and $\mathbf{w}$ is given in the Appendix A.1. The graphical illustration of our VMTL is shown in Fig. 1.

## 2.2 LEARNING TASK RELATEDNESS VIA GUMBEL-SOFTMAX PRIORS

The proposed variational multi-task inference framework enables the relationship among tasks to be explored by designing priors for both latent representations and classifiers. In Bayesian inference, priors serve as regularisation, which provides a principled way of sharing information across multiple tasks. We introduce the Gumbel-softmax prior for each task, which is a mixture of variational posteriors of other related tasks. In the case of latent representations, we define the prior by using other tasks posteriors of latent representations:

$$p(\mathbf{z}_t|\mathbf{x}_t) := \sum_{i \in \{1, \cdots, T\} \setminus t} \mathcal{A}_{ti} q(\mathbf{z}_i|\mathbf{m}_i; \phi), \qquad (9)$$

where $\mathbf{m}_i$ is the mean feature representation of samples from the same class in the $i$-th task. The mixing weight $\mathcal{A}_{ti}$ is defined as a binary value to indicate whether two tasks are correlated or not. To enable learning this binary $\mathcal{A}_{ti}$ with back propagation, we introduce the Gumbel-softmax technique (Jang et al., 2016):

$$\mathcal{A}_{ti} = \frac{\exp((\log \pi_{ti} + g_{ti})/\tau)}{\exp((\log \pi_{ti} + g_{ti})/\tau) + \exp((\log(1 - \pi_{ti}) + g'_{ti})/\tau)}, \qquad (10)$$

where $g_{ti}$ and $g'_{ti}$ are samples taken from a Gumbel distribution, using inverse transform sampling by drawing $u \sim \text{Uniform}(0, 1)$ and computing $g = -\log(-\log(u))$. $\pi_{ti}$ is the learnable parameter in the Gumbel-softmax technique, which denotes the probability of two tasks are correlated. Parameter $\tau$ is the softmax temperature.

Likewise, we specify the prior over classifier parameters $\mathbf{w}_t$ in the same way as in $\mathbf{z}_t$:

$$p(\mathbf{w}_t|\mathcal{D}_{1:T} \setminus \mathcal{D}_t) := \sum_{i \in \{1, \cdots, T\} \setminus t} \mathcal{A}_{ti}^{(\mathbf{w})} q(\mathbf{w}_i; \theta_i), \qquad (11)$$

where $\mathcal{A}_{ti}^{(\mathbf{w})}$ is obtained in a similar way as in Eq. (10). Note that we use different mixing weights in designing the priors for representations and classifiers, which gave better results than a shared one in our preliminary experiments. This is likely due to the fact that the representations and classifiers leverage different correlation patterns among different tasks.

It worth mentioning that the fundamental assumption in multi-task learning is that tasks are related and there is always positive transfer among them. Since tasks share the same label space in our setting, the case with only task interference would hardly happen. In case of only task interference, the KL term in Eq. (8) will degenerate to an $\ell_2$ regularization on the representation and classifiers.

## 2.3 AMORTIZED INFERENCE

We leverage an amortization technique (Gershman & Goodman, 2014), in which we amortize the computational cost of inferring the posterior of the latent representation, as done in VAEs (Kingma & Welling, 2013). In effect, the amortized inference can also be adopted to learn classifier parameters similar to the probabilistic prediction in few-shot learning (Gordon et al., 2018).

To this end, we design the variational posterior $q(\mathbf{w}_t|\mathcal{D}_t)$ in a context-independent manner such that each weight vector $\mathbf{w}_t^c$ depends only on samples from class $c$ of the current task $t$:

$$q(\mathbf{w}_t|\mathcal{D}_t) = \prod_{c=1}^{C} q(\mathbf{w}_t^c|\mathcal{D}_t^c) = \prod_{c=1}^{C} \mathcal{N}(\boldsymbol{\mu}_t^c, \text{diag}((\boldsymbol{\sigma}_t^c)^2)), \qquad (12)$$

where $\mathcal{D}^c$ are the samples from the $c$-th class, $C$ is the size of the shared label space, and each posterior is parameterized as a diagonal Gaussian distribution. The amortized inference of these posteriors is implemented by multi-layer perceptrons (MLPs) (Gordon et al., 2018; Edwards & Storkey, 2016; Kingma & Welling, 2013) and the parameters of the inference networks are jointly optimized in the end-to-end learning. For a given task, we use the amortized inference to generate the classifier weight for each specific class by using the mean feature representations in this class. Thus,

the weight for different classes are drawn from different distributions. In contrast to the amortized inference for latent representations, the amortized classifier inference enables the cost to be shared across classes, which reduces the overall cost. Therefore, it offers an effective way to handle a huge number of object classes and can still produce competitive performance even in the existence of the amortization gap (Cremer et al., 2018).

### 2.4 EMPIRICAL OBJECTIVE FUNCTION

In our implementation for the variational objective for multi-task learning Eq. (8), we adopt Monte Carlo sampling, and obtain the empirical objective function as follows:

$$
\hat{\mathcal{L}}_{\mathrm{VMTL}}(\boldsymbol{\theta}, \phi) = \frac{1}{T} \sum_{t=1}^{T} \Bigg[ \frac{1}{N_t} \sum_{n=1}^{N_t} \Big[ \frac{1}{ML} \sum_{\ell=1}^{L} \sum_{m=1}^{M} \big[ \log p(\mathbf{y}_t | \mathbf{z}_t^{n,(\ell)}, \mathbf{w}_t^{(m)}) \big]
$$
$$
- \mathbb{D}_{\mathrm{KL}} \big[ q(\mathbf{z}_t^n | \mathbf{x}_t^n; \phi) || p(\mathbf{z}_t^n | \mathbf{x}_t^n) \big] \Big] - \mathbb{D}_{\mathrm{KL}} \big[ q(\mathbf{w}_t; \theta_t) || p(\mathbf{w}_t | \mathcal{D}_{1:T} \backslash \mathcal{D}_t) \big] \Bigg], \tag{13}
$$

where $\mathbf{z}_t^{n,(\ell)} \sim q(\mathbf{z}_t^n | \mathbf{x}_t^n; \phi)$, $\mathbf{w}_t^{(m)} \sim q(\mathbf{w}_t; \theta)$. $L$ and $M$ are the number of Monte Carlo samples. In practice, $L$ and $M$ are set to 10, which performs well while being computationally efficient. We maximize this empirical function to optimize the model's parameters: the log-likelihood term is implemented as the cross-entropy loss and the $KL$ terms can be computed in closed forms. To sample from the variational posteriors, we adopt the reparameterization trick (Kingma & Welling, 2013). In the posterior inference of classifiers without amortization, we use the local reparameterization trick to reduce the gradient variance (Kingma et al., 2015). The priors $p(\mathbf{z}_t^n | \mathbf{x}_t^n)$ and $p(\mathbf{w}_t | \mathcal{D}_{1:T} \backslash \mathcal{D}_t)$ are implemented with Gumbel-softmax priors provided in Eq. (9) and Eq. (11), respectively. For amortized classifiers, the variational posterior $q(\mathbf{w}_t; \theta)$ is implemented using Eq. (12).

## 3 RELATED WORKS

Multi-task learning (Caruana, 1997) is a machine learning paradigm that aims to leverage shared knowledge from multiple related tasks to improve the generalization performance of all the tasks simultaneously. The core of multi-task learning is how to explore the task relationship, which has been extensively investigated in the literature.

Early works design feature-based or parameter-based regularizations to explore task relationships (Liu et al., 2009; Obozinski et al., 2010; Pong et al., 2010; Jalali et al., 2010; Kang et al., 2011; Jawanpuria et al., 2015; Long et al., 2017; Zhang et al., 2020). Obozinski et al. (2010) are the first to study the multi-task feature selection (MTFS) problem based on the $l_{2,1}$ norm. Liu et al. (2009) propose to use the $l_{\infty,1}$ norm with the objective function to select important features. Kang et al. (2011) design multiple task clusters, aiming to minimize the squared trace norm of the classifier parameters in each cluster.

Bayesian methods (Heskes, 2000; Bakker & Heskes, 2003; Yu et al., 2005; Xue et al., 2007; Guo et al., 2011; Titsias & Lázaro-Gredilla, 2011; Zhang & Yeung, 2012; Long et al., 2017) are developed for multi-task learning under probabilistic frameworks, where the regularization usually corresponds to a prior. Heskes (2000) proposes a Bayesian neural network for multi-task learning and analyse it with an empirical Bayesian framework. Yu et al. (2005) investigate Gaussian processes for multi-task learning assuming that all models are sampled from a common prior. Zhang & Yeung (2012) reformulate the $l_{p,q}$ norm regularizer as a matrix-variate generalized normal prior and utilize the prior information to explore task relations. Long et al. (2017) explores tensor normal distribution as priors of network parameters in different layers, which explicitly models the positive and negative relations across features and tasks. Xue et al. (2007) proposes a non-parametric hierarchical Bayesian model to avoid the high complexity of model parameters and are implemented with a deterministic inference method. Further, Qian et al. (2020) adopt a variational information bottleneck method (Alemi et al., 2016) with an uninformative prior distribution to improve the latent probabilistic representation. An important conclusion drawn by (Qian et al., 2020) is that, under adversarial attacks, variational latent representations are regularized and thereby expected to be more robust to noise than deterministic latent representations. The idea of conditioning priors on posteriors is also amenable to continual learning in that the posterior of the previous task can be used

as the prior of the current task to reduce catastrophic forgetting (Nguyen et al., 2017; Adel et al., 2019; Ebrahimi et al., 2019). In addition, Bragman et al. (2019) for the first time apply the Gumbel-Softmax to learn task relatedness in multi-task learning. Although these methods are also applicable to the data setting in this work and achieve encouraging improvements, they under-perform with very limited training data.

Deep learning has recently been explored for multi-task learning (Misra et al., 2016; Yang & Hospedales, 2016; Hashimoto et al., 2016; Ruder et al., 2017; Long et al., 2017; Meyerson & Miikkulainen, 2017; Chen et al., 2018; Maziarz et al., 2019) by designing different deep architectures to explore task relationships. Liu et al. (2019); Zhang et al. (2020) use a hard parameter-sharing encoder to extract the shared representations, while learning a task-specific decoder to obtain the pixel-level predictions. Based on soft parameter sharing, Misra et al. (2016) propose cross-stitch units to allow the model to leverage the shared knowledge from another task. Gao et al. (2020) follow the soft parameter sharing mechanism and incorporate neural architecture search into general-purpose multi-task learning. Meyerson & Miikkulainen (2017); Rosenbaum et al. (2017); Maziarz et al. (2019); Strezoski et al. (2019) develop flexible soft-ordering approaches to enable more effective sharing among tasks. Instead of learning the structure of sharing, Kendall et al. (2018) propose to weight multiple loss functions by considering the homoscedastic uncertainty of each task. Chen et al. (2018) present a gradient normalization algorithm that automatically balances training in deep multi-task models by dynamically tuning gradient magnitudes.

In our work, we address multi-task learning in a probabilistic inference framework by casting it as a variational Bayesian inference problem. We explore task relationships in a principled way by specifying priors conditioned on other tasks, which enables the model to share knowledge among related tasks for learning both representations and classifiers.

## 4    EXPERIMENTS

We conduct experiments on four benchmark datasets for multi-task learning with limited training data. The results demonstrate the benefits of variational Bayesian approximation to representations and classifiers for multi-task learning. Our VMTL achieves the best performance and consistently surpasses previous methods. We provide more experimental results in the Appendix B.

### 4.1    DATASETS

We evaluate the proposed VMTL under a challenging multi-task learning setting, where each task has limited training data and only a handful of related tasks that can be learned simultaneously to leverage the shared knowledge. To benchmark our model with previous methods, we conduct experiments on four benchmark datasets, where tasks are defined as classification problems from distinctive domains with a shared label space.

**Office-Home** (Venkateswara et al., 2017) contains images from four domains/tasks: Artistic (A), Clipart (C), Product (P) and Real-world (R). Each task contains images from 65 object categories collected in the office and home settings. There are about 15,500 images in total.

**Office-Caltech** (Gong et al., 2012) was constructed by selecting the ten categories shared between Office-31 (Saenko et al., 2010) and Caltech-256 datasets (Griffin et al., 2007). One task consists of data from Caltech-256 (C), and the other three tasks consist of data from Office-31 whose images were collected from three distinct domains/tasks, e.g., Amazon (A), Webcam (W) and DSLR (D). There are 8-151 samples per category per task, and 2,533 images in total.

**ImageCLEF** (Long et al., 2017), the benchmark for the ImageCLEF domain adaptation challenge, contains 12 common categories shared by four public datasets/tasks: Caltech-256 (C), ImageNet ILSVRC 2012 (I), Pascal VOC 2012 (P), and Bing (B). There are 2,400 images in total.

**DomainNet** (Peng et al., 2019), is a large-scale dataset with approximately 0.6 million images distributed among 345 categories. It contains 6 distinct domains: Clipart (C), Infograph (I), Painting (P), Quickdraw (Q), Real (R) and Sketch (S). This dataset provides an extremely challenging benchmark, which has never been introduced for multi-task learning

We follow the standard evaluation protocol (Zhang & Yeung, 2012; Zhang & Yang, 2017) for multi-task learning and randomly select 5%, 10%, and 20% of samples from each task as the training set

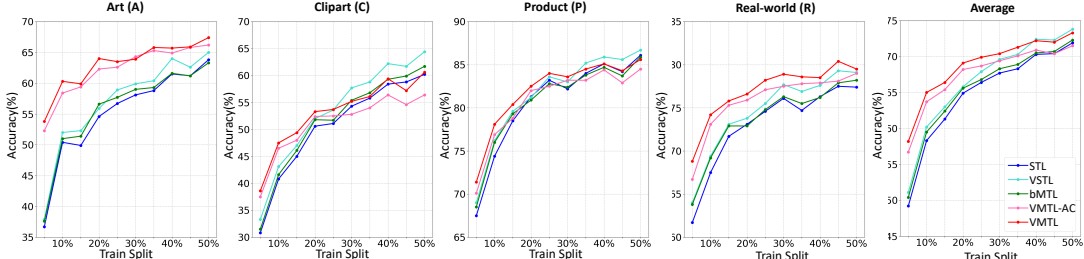

Fig. 2. The performance under different proportions of training data on the *Office-Home* dataset.

Table 1. Effectiveness of variational Bayesian approximation for representations and classifiers.

| Representation ($\mathbf{z}$) | Classifier ($\mathbf{w}$) | 5% | 10% | 20% |
|:---:|:---:|:---:|:---:|:---:|
| ✗ | ✗ | 50.4±0.1 | 59.5±0.1 | 65.6±0.1 |
| ✓ | ✗ | 51.3±0.2 | 60.4±0.1 | 66.7±0.2 |
| ✗ | ✓ | 56.8±0.1 | 63.7±0.1 | 68.5±0.1 |
| ✓ | ✓ | **58.2±0.2** | **65.0±0.0** | **69.1±0.1** |

Table 2. Performance comparison of VMTL with different priors. The detailed results on each individual task are provided in the Appendix B.3. Our Gumbel-softmax produces the best performance.

| Priors | 5% | 10% | 20% |
|:---:|:---:|:---:|:---:|
| Mean | 57.2±0.1 | 64.5±0.2 | 68.6±0.2 |
| Learnable weighted | 57.0±0.2 | 64.4±0.1 | 68.6±0.1 |
| Gumbel-softmax | **58.2±0.2** | **65.0±0.0** | **69.1±0.1** |

and use the remaining samples as the test set (Long et al., 2017). Note that when we use 5%, 10% and 20% labeled data for training, there are on average respectively 3, 6 and 12 samples per category per task. Hence, each task is provided with a limited amount of labeled data, which is insufficient for building reliable classifiers without overfitting. In order to maintain this data setting of limited training data, for the large-scale dataset **DomainNet**, we set the split to 1%, 2% and 4%, which results in an average of 3, 6 and 12 samples per category per task, respectively. This poses great challenges due to the huge number of object categories.

## 4.2 EXPERIMENTAL SETTINGS

**Implementation Details** Following the experimental settings in (Long et al., 2017), we remove the final classifier layer of VGGnet (Simonyan & Zisserman, 2014) and apply the remaining model to extract the feature representation $\mathbf{x}$ for each sample, from which we infer the latent representation $\mathbf{z}$ using amortized inference by MLPs (Kingma & Welling, 2013). In our experiments, the temperature is annealed using the same schedule as applied in (Jang et al., 2016): we start with a high temperature and gradually anneal it to a small but non-zero value. For the KL-divergence we use the annealing scheme from (Bowman et al., 2015), increasing the weight of the KL-divergence by a rate of 1e-6 per iteration. The dimension of the latent variable is set to 512. We adopt Adam optimizer (Kingma & Ba, 2014) with a learning rate of 1e-4 for training. All the results are obtained based on the 95% confidence interval from five runs.

**Comparison Methods** We compare our method with single-task learning (STL) as well as a variational version of STL (VSTL) implemented by introducing variational Bayesian inference to representations and classifiers without using task relationship (Details can be found in the Appendix A.2). We also define a basic multi-task learning (bMTL) model, which is a deep learning model with a shared feature extractor and task-specific classifiers. The bMTL serves as a baseline model to demonstrate the benefits of the probabilistic modeling of multi-task learning based on variational Bayesian inference. VMTL is our basic proposed method. VMTL-AC is our proposed method with amortized classifiers. We compare with multilinear relationship network (MRN) (Long et al., 2017), which holds the best performance among previous methods on the four benchmarks.

### 4.3 EXPERIMENTAL RESULTS

**Effectiveness in Handling Limited Data**  We conduct ablation studies to demonstrate the benefits of the proposed variational multi-task learning in exploring task relatedness. We conduct experiments under a large range of train-test splits from $5\%$ to $50\%$. The results on the Office-Home dataset for different tasks and the average accuracy of all tasks are illustrated in Fig. 2. More detailed experimental results are provided in the Appendix B.1. The performance advantage of our VMTL is larger in the settings with less training data ($\leq 20\%$), which demonstrates its effectiveness in handling challenging scenarios with very limited training data.

**Effectiveness of Variational Bayesian Approximation**  We investigate the effect of variational Bayesian inference for representations and classifiers, separately. We conduct these experiments on the Office-Home dataset. The results with different train-test splits are shown in Table 1. More detailed experimental results are provided in the Appendix B.2. As can be seen, both variational Bayesian representations and classifiers can benefit performance. This benefit becomes more significant when training data is very limited, which indicates the effectiveness of leveraging shared knowledge by conditioning priors on related tasks in our VMTL. In addition, in Fig. 2, the VSTL is shown to be a strong learning model, which again demonstrates the benefits of variational Bayesian representations and classifiers compared to STL.

Table 3.  Performance comparison of different methods on the *Office-Home* dataset for multiple tasks: Artistic (A), Clipart (C), Product (P) and Real-world (R).

| Methods | 5% | | | | | 10% | | | | |
|---|---|---|---|---|---|---|---|---|---|---|
| | A | C | P | R | Avg. | A | C | P | R | Avg. |
| STL | 36.7 | 30.8 | 67.5 | 61.7 | 49.2 | 50.4 | 40.8 | 74.4 | 67.5 | 58.3 |
| VSTL | 37.9 | 33.3 | 69.0 | 64.0 | 51.1 | 52.0 | 43.1 | 76.2 | 69.4 | 60.2 |
| MRN | 53.3 | 36.4 | 70.5 | 67.7 | 57.0 | 59.9 | 42.7 | 76.3 | 73.0 | 63.0 |
| bMTL | 37.6±0.4 | 31.5±0.3 | 68.5±0.2 | 63.8±0.2 | 50.4±0.1 | 51.0±0.2 | 41.6±0.1 | 76.0±0.3 | 69.2±0.3 | 59.5±0.1 |
| **VMTL-AC** | 52.3±0.4 | 37.5±0.5 | 70.1±0.3 | 66.7±0.2 | 56.7±0.2 | 58.4±0.5 | 46.5±0.3 | 76.9±0.2 | 73.1±0.3 | 63.7±0.1 |
| **VMTL** | 53.8±0.6 | 38.6±0.2 | 71.4±0.3 | 68.8±0.2 | **58.2±0.2** | 60.3±0.5 | 47.5±0.2 | 78.1±0.2 | 74.2±0.1 | **65.0±0.0** |

Table 4.  Performance comparison of different methods on the *Office-Caltech* dataset for multiple tasks: Amazon (A), Webcam (W), DSLR (D) and Caltech-256 (C).

| Methods | 5% | | | | | 10% | | | | |
|---|---|---|---|---|---|---|---|---|---|---|
| | A | W | D | C | Avg. | A | W | D | C | Avg. |
| STL | 87.4 | 87.9 | 96.4 | 82.8 | 88.6 | 92.8 | 97.7 | 87.8 | 84.3 | 90.7 |
| VSTL | 88.3 | 89.1 | 97.0 | 81.4 | 89.0 | 93.1 | 96.6 | 90.0 | 84.5 | 91.1 |
| MRN | 92.7 | 94.3 | 97.1 | 89.2 | 93.4 | 95.0 | 98.1 | 95.0 | 91.3 | 94.8 |
| bMTL | 90.0±0.7 | 89.4±0.8 | 95.0±1.1 | 83.5±0.5 | 89.5±0.3 | 93.6±0.1 | 97.0±0.6 | 92.1±0.7 | 86.3±0.4 | 92.3±0.2 |
| **VMTL-AC** | 93.2±0.3 | 95.0±0.3 | 96.1±0.3 | 89.7±0.5 | 93.5±0.1 | 94.8±0.3 | 96.8±0.4 | 97.7±0.3 | 90.1±0.3 | 94.9±0.2 |
| **VMTL** | 93.8±0.3 | 95.5±0.4 | 96.4±0.4 | 90.0±0.3 | **93.9±0.2** | 95.5±0.1 | 97.0±0.1 | 97.9±0.3 | 91.0±0.1 | **95.3±0.1** |

**Effectiveness of Gumbel-Softmax Priors**  The introduced Gumbel-softmax prior provides an effective way to learn data-driven task relationships. To demonstrate their effectiveness, we compare with several alternatives, including the mean and the learnable weighted posteriors of other tasks. The comparison results are shown in Table 2. The proposed Gumbel-softmax priors perform the best, consistently surpassing other alternatives. It is also worth noting that the advantage of Gumbel-softmax priors is ever larger for very limited training data, e.g., $5\%$, a challenging scenario where it is crucial to leverage task relatedness.

**Comparison with other methods**  The comparison results on the small-scale datasets *Office-Home*, *Office-Caltech*,*ImageCLEF datasets* and large-scale *DomainNet* datasets are shown in Tables 3, 4, 5 and 6, respectively. The average accuracy of all tasks is used for performance measurement. The best results of average accuracy are marked in bold, while the second-best by underline. Due to the space limitation, we show the experimental results with more data accessible in Table 15 and Table 16 of the Appendix B.4. A comprehensive comparison with more other methods is provided in the Appendix B.4.

Our VMTL consistently achieves the best performance on all small-scale and large-scale datasets with all the defined train-test split settings. It is worth highlighting that on the most challenging setting of $5\%$ training data, our VMTL shows a large performance advantage over compared methods.

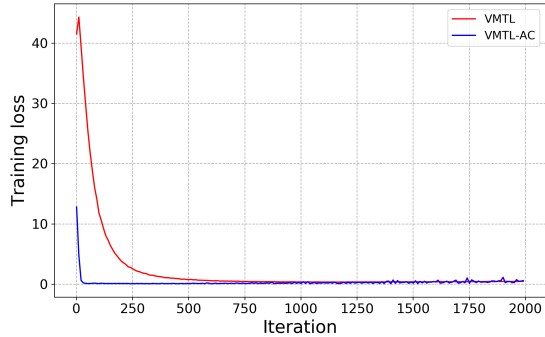

Fig. 3. The illustration of training loss with iterations. VMTL-AC converges faster than VMTL, which demonstrates its computation benefit by amortized learning.

Table 5. Performance comparison of different methods on the *ImageCLEF* dataset for multiple tasks: Caltech-256 (C), ImageNet ILSVRC 2012 (I), Pascal VOC 2012 (P), and Bing (B).

| Methods | 5% | | | | | 10% | | | | |
|---|---|---|---|---|---|---|---|---|---|---|
| | C | I | P | B | Avg. | C | I | P | B | Avg. |
| STL | 85.4 | 71.4 | 57.7 | 36.0 | 62.6 | 88.9 | 77.8 | 64.3 | 47.6 | 69.7 |
| VSTL | 87.0 | 73.2 | 60.5 | 39.0 | 64.9 | 89.6 | 79.1 | 66.6 | 48.0 | 70.8 |
| MRN | 90.1 | 76.5 | 72.8 | 54.9 | 73.7 | 93.3 | 83.2 | 70.4 | 56.3 | 75.8 |
| bMTL | 88.3±0.5 | 73.2±0.5 | 61.2±0.9 | 40.0±0.8 | 65.7±0.4 | 90.6±0.8 | 79.3±0.2 | 66.3±0.5 | 51.9±0.6 | 72.0±0.3 |
| **VMTL-AC** | 89.1±0.4 | 81.4±0.9 | 71.2±0.4 | 56.5±1.1 | 74.5±0.5 | 92.3±0.4 | 83.9±0.8 | 71.8±0.8 | 58.9±0.8 | 76.7±0.2 |
| **VMTL** | 91.1±0.3 | 83.2±0.6 | 71.4±0.4 | 58.3±0.8 | **76.0±0.2** | 93.7±0.4 | 86.5±0.4 | 71.8±0.4 | 59.5±0.6 | **77.9±0.2** |

Table 6. Performance comparison of different methods on the large-scaled dataset *DomainNet* for multiple tasks: Clipart (C), Infograph (I), Painting (P), Quickdraw (Q), Real (R) and Sketch (S).

| Methods | 1% | | | | | | |
|---|---|---|---|---|---|---|---|
| | C | I | P | Q | R | S | Avg. |
| STL | 15.0 | 4.0 | 19.7 | 7.5 | 50.6 | 9.6 | 17.7 |
| VSTL | 18.9 | 5.4 | 23.5 | 15.2 | 54.5 | 12.2 | 21.6 |
| bMTL | 18.3±0.1 | 5.2±0.2 | 22.3±0.1 | 15.0±0.1 | 53.3±0.1 | 11.8±0.2 | 21.0±0.1 |
| **VMTL-AC** | 18.6±0.3 | 5.7±0.2 | 23.0±0.2 | 12.7±0.1 | 51.6±0.2 | 12.5±0.2 | 20.7±0.1 |
| **VMTL** | 24.8±0.1 | 8.5±0.1 | 29.9±0.0 | 12.7±0.2 | 56.9±0.1 | 16.9±0.1 | **25.0±0.0** |

Specifically, on ImageCLEF, under the 5% setting, our VMTL surpasses the second best by a phenomenal margin up to 2.3%. This demonstrates the effectiveness of VMTL in exploring relatedness to improve the performance of each task. In addition, our VMTL-AC can also produce comparable performance and is better than most previous methods. It is worth mentioning that VMTL-AC demonstrates computation advantages with faster convergence compared to VMTL due to the amortized learning as shown in Fig. 3. Besides, we found that VMTL-AC demonstrates good robustness against adversarial attacks. This could be due to that amortized learning applies the mean feature representations to generate classifiers, which is more robust to attacks. The detailed discussions are given in Appendix B.5. Finally, the improvement of VSTL over STL also indicates the benefits of variational Bayesian approximation for representations and classifiers.

## 5 CONCLUSION

In this paper, we address the multi-task learning problem and tackle a challenging setting where each task has a very limited amount of training data, with only a handful of related tasks. To this end, we develop *variational multi-task learning* - VMTL, a general probabilistic inference framework for simultaneously learning multiple tasks. We cast multi-task learning as a variational inference problem, which enables task relationships to be explored in a principled way by specifying priors. Specifically, we introduce the Gumbel-softmax priors, which offer an effective way to learn the task relatedness in a data-driven manner for each task. We evaluate VMTL on four benchmark datasets for multi-task learning. Results demonstrate that our VMTL consistently achieves better or comparable performance with state-of-the-art multi-task learning approaches.

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

## A DERIVATION

### A.1 DERIVATION OF EVIDENCE LOWER BOUND FOR MULTI-TASK LEARNING

We provide a derivation of the evidence lower bound with $\mathbf{z}$ and $\mathbf{w}$ jointly. The log-likelihood of task $t$ is conditioned on the data from other related tasks.

$$\log p(\mathbf{y}_t|\mathbf{x}_t, \mathcal{D}_{1:T}\backslash\mathcal{D}_t) = \log \int \int p(\mathbf{y}_t, \mathbf{z}_t, \mathbf{w}_t, |\mathbf{x}_t, \mathcal{D}_{1:T}\backslash\mathcal{D}_t)d\mathbf{w}_t d\mathbf{z}_t \tag{14}$$

Under the assumption that $\mathbf{w}_t$ and $\mathbf{z}_t$ are conditionally independent, we therefore obtain

$$\begin{aligned}
&\log p(\mathbf{y}_t|\mathbf{x}_t, \mathcal{D}_{1:T}\backslash\mathcal{D}_t) \\
&= \log \int \int p(\mathbf{y}_t|\mathbf{z}_t, \mathbf{w}_t)p(\mathbf{z}_t|\mathbf{x}_t)p(\mathbf{w}_t|\mathcal{D}_{1:T}\backslash\mathcal{D}_t)d\mathbf{w}_t d\mathbf{z}_t \\
&= \log \int \big[ \int p(\mathbf{y}_t|\mathbf{z}_t, \mathbf{w}_t)p(\mathbf{z}_t|\mathbf{x}_t)d\mathbf{z}_t \big] p(\mathbf{w}_t|\mathcal{D}_{1:T}\backslash\mathcal{D}_t)d\mathbf{w}_t \\
&= \log \int \frac{\big[ \int p(\mathbf{y}_t|\mathbf{z}_t, \mathbf{w}_t)p(\mathbf{z}_t|\mathbf{x}_t)d\mathbf{z}_t \big] p(\mathbf{w}_t|\mathcal{D}_{1:T}\backslash\mathcal{D}_t)q(\mathbf{w}_t)}{q(\mathbf{w}_t)}d\mathbf{w}_t \\
&\geq -\mathbb{KL}[q(\mathbf{w}_t)||p(\mathbf{w}_t|\mathcal{D}_{1:T}\backslash\mathcal{D}_t)] + \mathbb{E}_{q(\mathbf{w}_t)}\big[ \log \int p(\mathbf{y}_t|\mathbf{z}_t, \mathbf{w}_t)p(\mathbf{z}_t|\mathbf{x}_t)d\mathbf{z}_t \big] \\
&\geq -\mathbb{KL}[q(\mathbf{w}_t)||p(\mathbf{w}_t|\mathcal{D}_{1:T}\backslash\mathcal{D}_t)] + \mathbb{E}_{q(\mathbf{w}_t)}\big[ \log \int \frac{p(\mathbf{y}_t|\mathbf{z}_t, \mathbf{w}_t)p(\mathbf{z}_t|\mathbf{x}_t)q(\mathbf{z}_t|\mathbf{x}_t)}{q(\mathbf{z}_t|\mathbf{x}_t)}d\mathbf{z}_t \big] \\
&\geq -\mathbb{KL}[q(\mathbf{w}_t)||p(\mathbf{w}_t|\mathcal{D}_{1:T}\backslash\mathcal{D}_t)] - \mathbb{KL}[q(\mathbf{z}_t|\mathbf{x}_t)||p(\mathbf{z}_t|\mathbf{x}_t)] + \mathbb{E}_{q(\mathbf{w}_t)}\mathbb{E}_{q(\mathbf{z}_t|\mathbf{x}_t)}[\log p(\mathbf{y}_t|\mathbf{z}_t, \mathbf{w}_t)]
\end{aligned} \tag{15}$$

### A.2 DERIVATION OF EVIDENCE LOWER BOUND FOR SINGLE-TASK LEARNING

Generally, the proposed Bayesian inference framework which infers the posteriors of presentations $\mathbf{z}$ and classifiers $\mathbf{w}$ jointly can be widely applied in other research fields. To be simple, we introduce a variational version of single-task learning (VSTL), and provide the derivation of its evidence lower bound. It is worth noting that single-task learning does not share knowledge among tasks. Thus, the log-likelihood for single-task learning is not conditioned on the data from other related tasks.

$$\begin{aligned}
\log p(\mathbf{y}|\mathbf{x}) &= \log \int \int p(\mathbf{y}, \mathbf{z}, \mathbf{w}, |\mathbf{x})d\mathbf{w}d\mathbf{z} \\
&= \log \int \int p(\mathbf{y}|\mathbf{z}, \mathbf{w})p(\mathbf{w})p(\mathbf{z}|\mathbf{x})d\mathbf{w}d\mathbf{z} \\
&= \log \int \int p(\mathbf{y}|\mathbf{z}, \mathbf{w})p(\mathbf{z}|\mathbf{x})d\mathbf{z}p(\mathbf{w})d\mathbf{w} \\
&= \log \int \frac{[\int p(\mathbf{y}|\mathbf{z}, \mathbf{w})p(\mathbf{z}|\mathbf{x})d\mathbf{z}]p(\mathbf{w})q(\mathbf{w})}{q(\mathbf{w})}d\mathbf{w} \\
&\geq -\mathbb{KL}(q(\mathbf{w})||p(\mathbf{w})) + \mathbb{E}_{q(\mathbf{w})}[\log \int p(\mathbf{y}|\mathbf{z}, \mathbf{w})p(\mathbf{z}|\mathbf{x})d\mathbf{z}] \\
&\geq -\mathbb{KL}[q(\mathbf{w})||p(\mathbf{w})] + \mathbb{E}_{q(\mathbf{w})}[\log \int \frac{p(\mathbf{y}|\mathbf{z}, \mathbf{w})p(\mathbf{z}|\mathbf{x})q(\mathbf{z}|\mathbf{x})}{q(\mathbf{z}|\mathbf{x})}d\mathbf{z}] \\
&\geq -\mathbb{KL}[q(\mathbf{w})||p(\mathbf{w})] - \mathbb{KL}[q(\mathbf{z}|\mathbf{x})||p(\mathbf{z}|\mathbf{x})] + \mathbb{E}_{q(\mathbf{w})}\mathbb{E}_{q(\mathbf{z}|\mathbf{x})}[\log p(\mathbf{y}|\mathbf{z}, \mathbf{w})]
\end{aligned} \tag{16}$$

Usually, the approximate posteriors $q(\mathbf{w})$ and $q(\mathbf{z}|\mathbf{x})$ are defined as a fully-factorized Gaussian distribution. Due to lack of extracted information offered by other related tasks, the priors $p(\mathbf{w})$ and $p(\mathbf{z}|\mathbf{x})$ are set to a standard Gaussian distribution, as applied in (Sohn et al., 2015; Kingma et al., 2015; Molchanov et al., 2017).

# B EXTRA EXPERIMENTAL RESULTS

## B.1 EFFECTIVENESS IN HANDLING LIMITED DATA

We further provide detailed information in Table 7 about average accuracy in Fig. 2. Our proposed probabilistic models, i.e., VMTL and VMTL-AC outperform the deterministic baseline multi-task learning model (bMTL), which demonstrates the benefits of our proposed variational Bayesian framework. Given a limited amount of training data, STL and VSTL can not train a proper model for each task. As the training data decreases, our methods based on the variational Bayesian framework are able to better handle this challenging case by incorporating the shared knowledge into the prior of each tasks. The best results of average accuracy are marked in bold, while the second-best by underline.

Table 7. Performance of average accuracy under different proportions of training data on *Office-Home*.

| Methods | 5% | 10% | 15% | 20% | 25% | 30% | 35% | 40% | 45% | 50% |
|---------|-----|------|------|------|------|------|------|------|------|------|
| STL | 49.2 | 58.3 | 61.3 | 64.9 | 66.4 | 67.7 | 68.3 | 70.3 | 70.4 | 71.9 |
| VSTL | 51.1 | 60.2 | 63.0 | 65.8 | 67.9 | 69.6 | 70.3 | **72.4** | **72.3** | **73.8** |
| bMTL | 50.4±0.1 | 59.5±0.1 | 62.4±0.1 | 65.6±0.1 | 66.8±0.1 | 68.3±0.2 | 68.9±0.1 | 70.5±0.1 | 70.7±0.2 | 72.3±0.2 |
| VMTL-AC | 56.6±0.2 | 63.7±0.1 | 65.4±0.2 | 68.1±0.1 | 68.7±0.1 | 69.4±0.2 | 70.0±0.1 | 70.9±0.1 | 70.4±0.1 | 71.5±0.1 |
| VMTL | **58.2±0.2** | **65.0±0.0** | **66.4±0.1** | **69.1±0.1** | **69.9±0.1** | **70.4±0.1** | **71.3±0.1** | 72.2±0.1 | 72.0±0.1 | 73.3±0.1 |

## B.2 EFFECTIVENESS VARIATIONAL BAYESIAN APPROXIMATION

The comparison results on performance of Bayesian approximation for representations **z** and classifiers **w** on the Office-Home, Office-Caltech and ImageCLEF datasets are shown in Tables 8, 9 and 10, respectively. Both variational Bayesian representations and classifiers can benefit performance. And we find that Bayesian classifiers in the variational inference framework contribute more to the performance than Bayesian representations. It is likely due to the fact that Bayesian classifiers can better improve the model's discriminative ability. Our method jointly infers the posteriors over feature representations and classifiers in a Bayesian framework, which consistently outperforms its variants on three benchmarks.

## B.3 EFFECTIVENESS OF GUMBEL-SOFTMAX PRIORS

The performance comparison of the proposed VMTL with different priors on the Office-Home, Office-Caltech and ImageCLEF datasets is shown in Tables 11, 12 and 13, respectively. "Mean" denotes that the prior of the current task is the mean of variational posteriors of other related tasks. "Learnable weighted" denotes that weights of mixing the variational posteriors of other related tasks are learnable. Our Gumbel-softmax Priors apply the Gumbel-softmax technique to learn the mixing weights, which introduces uncertainty to the relationships among tasks in order to explore sufficient transferable information from other tasks. In the three datasets, our designed priors outperform other methods consistently.

## B.4 A COMPREHENSIVE COMPARISON WITH OTHER METHODS

The comprehensive comparison with state-of-the-art methods, including multi-task feature learning (MTFL) (Argyriou et al., 2007), robust multi-task learning (RMTL) (Chen et al., 2011), multi-task relationship learning (MTRL) Zhang & Yeung (2012), deep multi-task learning with tensor factorization (DMRL-TF) (Yang & Hospedales, 2016) and multilinear relationship network (MRN) (Long et al., 2017) is shown in Table 14. The results of the above state-of-the-art methods are taken from paper (Long et al., 2017). The results of three datasets under the 20% train-test split and the results of *DomainNet* under the 2% and 4% train-test split are is provided in Table 15 and Table 16, respectively. The proposed VMTL consistently achieves the best performance on all datasets with all train-test split settings. VMTL with amortized classifiers (VMTL-AC) can produce competitive performance better than most of previous methods.

## B.5    ROBUSTNESS OF OUR METHODS

We conduct some experiments to show the robustness of our methods against adversarial attacks. In our experiments, the adversarial attack is implemented by the fast gradient sign method (Goodfellow et al., 2014) where $\epsilon$ denotes the noise level. We evaluate our proposed VMTL, VMTL-AC, and the basic multi-task learning(bMTL) on the Office-home dataset. As shown in Fig. 4, under different noise levels, VMTL outperforms bMTL. As the noise level increases, the variant of our method VMTL-AC is more robust and significantly outperforms the baseline multi-task learning model.

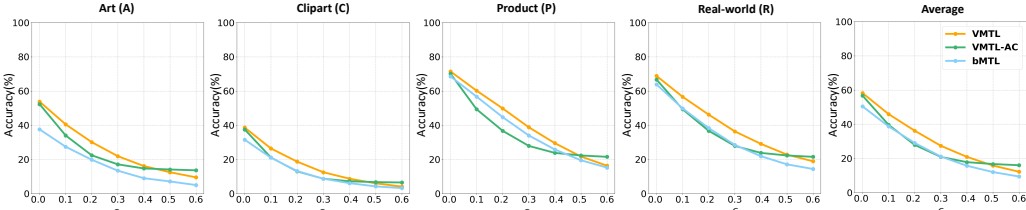

Fig. 4.   The performance for each task under different noise level on the *Office-Home* dataset.

Table 8. Detailed results on performance of Bayesian approximation for representation **z** and classifier **w** on *Office-Home*.

| z | w | 5% | | | | | 10% | | | | | 20% | | | | |
|---|---|---|---|---|---|---|---|---|---|---|---|---|---|---|---|---|
| | | A | C | P | R | Avg. | A | C | P | R | Avg. | A | C | P | R | Avg. |
| × | × | 37.6±0.4 | 31.5±0.3 | 68.5±0.2 | 63.8±0.2 | 50.4±0.1 | 51.0±0.2 | 41.6±0.1 | 76.0±0.3 | 69.2±0.3 | 59.5±0.1 | 56.6±0.3 | 51.8±0.5 | 80.9±0.3 | 72.9±0.4 | 65.6±0.1 |
| ✓ | × | 39.2±0.3 | 32.7±0.3 | 69.1±0.2 | 64.4±0.3 | 51.4±0.2 | 52.7±0.4 | 42.5±0.2 | 76.6±0.3 | 70.2±0.4 | 60.5±0.1 | 57.8±0.5 | 53.2±0.3 | 81.8±0.2 | 74.1±0.2 | 66.7±0.2 |
| × | ✓ | 51.6±0.3 | 37.8±0.1 | 70.7±0.4 | 66.9±0.2 | 56.8±0.1 | 57.6±0.3 | 47.2±0.2 | 77.4±0.3 | 72.5±0.1 | 63.7±0.1 | 62.5±0.4 | 53.5±0.3 | 82.0±0.2 | 76.0±0.3 | 68.5±0.1 |
| ✓ | ✓ | 53.8±0.6 | 38.6±0.2 | 71.4±0.3 | 68.8±0.2 | **58.2±0.2** | 60.3±0.5 | 47.5±0.2 | 78.1±0.2 | 74.2±0.1 | **65.0±0.0** | 64.0±0.3 | 53.3±0.3 | 82.5±0.2 | 76.6±0.3 | **69.1±0.1** |

Table 9. Detailed results on performance of Bayesian approximation for representation **z** and classifier **w** on *Office-Caltech*.

| z | w | 5% | | | | | 10% | | | | | 20% | | | | |
|---|---|---|---|---|---|---|---|---|---|---|---|---|---|---|---|---|
| | | A | W | D | C | Avg. | A | W | D | C | Avg. | A | W | D | C | Avg. |
| × | × | 90.0±0.7 | 89.4±0.8 | 95.0±1.1 | 83.5±0.5 | 89.5±0.3 | 93.6±0.1 | 97.0±0.6 | 92.1±0.7 | 86.3±0.4 | 92.3±0.2 | 95.0±0.2 | 94.5±0.7 | 96.0±1.2 | 86.8±0.2 | 93.1±0.1 |
| ✓ | × | 91.0±0.3 | 91.3±0.7 | 97.9±0.7 | 84.5±0.3 | 91.2±0.4 | 94.4±0.1 | 97.0±0.2 | 94.3±0.7 | 87.1±0.4 | 93.2±0.2 | 95.7±0.1 | 94.9±0.2 | 95.2±0.8 | 87.5±0.3 | 93.3±0.2 |
| × | ✓ | 93.2±0.1 | 95.5±0.3 | 95.7±0.5 | 89.6±0.2 | 93.5±0.1 | 94.8±0.2 | 97.3±0.4 | 97.8±0.4 | 90.6±0.3 | 95.1±0.1 | 95.6±0.2 | 96.6±0.3 | 99.2±0.3 | 89.8±0.4 | 95.3±0.2 |
| ✓ | ✓ | 93.8±0.3 | 95.5±0.4 | 96.4±0.4 | 90.0±0.3 | **93.9±0.2** | 95.5±0.1 | 97.0±0.1 | 97.9±0.3 | 91.0±0.1 | **95.3±0.1** | 95.6±0.1 | 95.8±0.4 | 99.2±0.6 | 90.6±0.5 | **95.3±0.1** |

Table 10. Detailed results on performance of Bayesian approximation for representation **z** and classifier **w** on *ImageCLEF*.

| z | w | 5% | | | | | 10% | | | | | 20% | | | | |
|---|---|---|---|---|---|---|---|---|---|---|---|---|---|---|---|---|
| | | C | I | P | B | Avg. | C | I | P | B | Avg. | C | I | P | B | Avg. |
| × | × | 88.3±0.5 | 73.2±0.5 | 61.2±0.9 | 40.0±0.8 | 65.7±0.4 | 90.6±0.8 | 79.3±0.2 | 66.3±0.5 | 51.9±0.6 | 72.0±0.3 | 92.9±0.5 | 86.5±0.2 | 71.9±0.8 | 56.0±0.8 | 76.8±0.3 |
| ✓ | × | 89.0±0.4 | 73.5±0.8 | 62.6±0.7 | 42.4±0.8 | 66.9±0.2 | 91.3±0.6 | 79.8±0.3 | 68±0.8 | 51.5±0.4 | 72.6±0.3 | 93.5±0.2 | 88.1±0.5 | 74.0±0.2 | 57.5±0.8 | 78.3±0.2 |
| × | ✓ | 91.4±1.1 | 81.9±0.5 | 71.8±0.6 | 58.4±0.5 | 75.9±0.2 | 93.0±0.6 | 86.3±0.4 | 72.0±0.6 | 60.0±0.6 | 77.8±0.3 | 93.8±0.5 | 88.1±0.5 | 77.1±0.8 | 59.0±0.4 | 79.5±0.2 |
| ✓ | ✓ | 91.1±0.3 | 83.2±0.6 | 71.4±0.4 | 58.3±0.8 | **76.0±0.2** | 93.7±0.4 | 86.5±0.4 | 71.8±0.4 | 59.5±0.6 | **77.9±0.2** | 94.0±0.2 | 89.7±0.3 | 77.9±0.4 | 59.7±0.5 | **80.3±0.2** |

Table 11. Detailed results on performance of VMTL with different priors on *Office-Home*.

| Priors | 5% | | | | | 10% | | | | | 20% | | | | |
|---|---|---|---|---|---|---|---|---|---|---|---|---|---|---|---|
| | A | C | P | R | Avg. | A | C | P | R | Avg. | A | C | P | R | Avg. |
| Mean | 52.0±0.4 | 38.0±0.2 | 70.6±0.1 | 68.0±0.3 | 57.2±0.1 | 59.7±0.4 | 47.0±0.3 | 77.9±0.2 | 73.5±0.3 | 64.5±0.2 | 63.0±0.5 | 53.0±0.4 | 82.0±0.2 | 76.5±0.2 | 68.6±0.2 |
| Learnable weighted | 51.4±0.4 | 38.0±0.2 | 70.7±0.2 | 67.7±0.2 | 57.0±0.2 | 59.8±0.3 | 47.2±0.3 | 77.8±0.1 | 72.7±0.3 | 64.4±0.1 | 63.1±0.4 | 53.3±0.3 | 82.2±0.3 | 75.9±0.3 | 68.6±0.1 |
| Gumbel-Softmax | 53.8±0.6 | 38.6±0.2 | 71.4±0.3 | 68.8±0.2 | **58.2±0.2** | 60.3±0.5 | 47.5±0.2 | 78.1±0.2 | 74.2±0.1 | **65.0±0.0** | 64.0±0.3 | 53.3±0.3 | 82.5±0.2 | 76.6±0.3 | **69.1±0.1** |

Table 12. Detailed results on performance of VMTL with different priors on *Office-Caltech*.

| Priors | 5% | | | | | 10% | | | | | 20% | | | | |
|---|---|---|---|---|---|---|---|---|---|---|---|---|---|---|---|
| | A | W | D | C | Avg. | A | W | D | C | Avg. | A | W | D | C | Avg. |
| Mean | 93.8±0.5 | 95.8±0.5 | 97.1±0.5 | 88.5±0.4 | 93.8±0.2 | 95.3±0.1 | 97.7±0.2 | 97.1±0.3 | 90.4±0.2 | 95.1±0.0 | 95.7±0.3 | 96.2±0.4 | 98.4±0.6 | 90.0±0.3 | 95.1±0.1 |
| Learnable weighted | 92.9±0.4 | 94.7±0.4 | 96.4±0.4 | 89.9±0.3 | 93.5±0.1 | 95.5±0.1 | 97.0±0.3 | 97.9±0.0 | 90.2±0.2 | 95.2±0.1 | 95.8±0.2 | 94.5±0.3 | 99.2±0.9 | 90.7±0.6 | 95.1±0.1 |
| Gumbel-softmax | 93.8±0.3 | 95.5±0.4 | 96.4±0.4 | 90.0±0.3 | **93.9±0.2** | 95.5±0.1 | 97.0±0.1 | 97.9±0.3 | 91.0±0.1 | **95.3±0.1** | 95.6±0.1 | 95.8±0.4 | 99.2±0.6 | 90.6±0.5 | **95.3±0.1** |

Table 13. Detailed results on performance of VMTL with different priors on *ImageCLEF*.

| Priors | 5% | | | | | 10% | | | | | 20% | | | | |
|---|---|---|---|---|---|---|---|---|---|---|---|---|---|---|---|
| | C | I | P | B | Avg. | C | I | P | B | Avg. | C | I | P | B | Avg. |
| Mean | 89.5±0.6 | 83.0±0.7 | 71.4±0.1 | 58.8±0.4 | 75.7±0.2 | 93.5±0.4 | 86.5±0.3 | 71.7±0.4 | 60.2±0.4 | 78.0±0.2 | 93.1±0.5 | 89.6±0.5 | 78.8±0.8 | 58.8±0.7 | 80.1±0.2 |
| Learnable weighted | 90.7±0.6 | 81.8±0.4 | 71.9±0.7 | 57.4±0.8 | 75.5±0.4 | 93.7±0.3 | 86.1±0.4 | 71.9±0.8 | 59.3±0.6 | 77.8±0.2 | 93.3±0.4 | 89.6±0.3 | 77.7±0.6 | 59.2±0.3 | 79.9±0.3 |
| Gumbel-softmax | 91.1±0.3 | 83.2±0.6 | 71.4±0.4 | 58.3±0.8 | **76.0±0.2** | 93.7±0.4 | 86.5±0.4 | 71.8±0.4 | 59.5±0.6 | **77.9±0.2** | 94.0±0.2 | 89.7±0.3 | 77.9±0.4 | 59.7±0.5 | **80.3±0.2** |

Table 14. Performance comparison of different methods on the *Office-Home* dataset for multiple tasks: Artistic (A), Clipart (C), Product (P) and Real-world (R).

| Methods | 5% | | | | | 10% | | | | | 20% | | | | |
|---|---|---|---|---|---|---|---|---|---|---|---|---|---|---|---|
| | A | C | P | R | Avg. | A | C | P | R | Avg. | A | C | P | R | Avg. |
| MTFL | 40.1 | 30.4 | 61.5 | 59.5 | 47.9 | 50.3 | 35.0 | 66.3 | 65.0 | 54.2 | 55.2 | 38.8 | 69.1 | 70.0 | 58.3 |
| RMTL | 42.3 | 32.8 | 62.3 | 60.6 | 49.5 | 49.7 | 34.6 | 65.9 | 64.6 | 53.7 | 55.2 | 39.2 | 69.6 | 70.5 | 58.6 |
| MTRL | 42.7 | 33.3 | 62.9 | 61.3 | 50.1 | 51.6 | 36.3 | 67.7 | 66.3 | 55.5 | 55.8 | 39.9 | 70.2 | 71.2 | 59.3 |
| DMTL-TF | 49.2 | 34.5 | 67.1 | 62.9 | 53.4 | 57.2 | 42.3 | 73.6 | 69.9 | 60.8 | 58.3 | 56.1 | 79.3 | 72.1 | 66.5 |
| MRN | 53.3 | 36.4 | 70.5 | 67.7 | 57.0 | 59.9 | 42.7 | 76.3 | 73.0 | 63.0 | 58.5 | 55.6 | 80.7 | 72.8 | 66.9 |
| **VMTL-AC** | 52.3±0.4 | 37.5±0.5 | 70.1±0.3 | 66.7±0.2 | 56.7±0.2 | 58.4±0.5 | 46.5±0.3 | 76.9±0.2 | 73.1±0.3 | 63.7±0.1 | 62.3±0.1 | 52.4±0.3 | 82.0±0.2 | 75.9±0.4 | 68.2±0.1 |
| **VMTL** | 53.8±0.6 | 38.6±0.2 | 71.4±0.3 | 68.8±0.2 | **58.2±0.2** | 60.3±0.5 | 47.5±0.2 | 78.1±0.2 | 74.2±0.1 | **65.0±0.0** | 64.0±0.3 | 53.3±0.2 | 82.5±0.2 | 76.6±0.3 | **69.1±0.1** |

Table 15. Performance comparison of different methods on the three dataset with 20% train-split: *Office-Home, Office-Caltech, ImageCLEF*

| Methods | Office-Home (20%) | | | | | Office-Caltech(20%) | | | | | ImageCLEF(20%) | | | | |
|---|---|---|---|---|---|---|---|---|---|---|---|---|---|---|---|
| | A | C | P | R | Avg. | A | W | D | C | Avg. | C | I | P | B | Avg. |
| STL | 54.6 | 50.6 | 81.3 | 73.1 | 64.9 | 94.9 | 92.8 | 95.2 | 86.7 | 92.4 | 92.9 | 84.6 | 72.5 | 54.6 | 76.2 |
| VSTL | 55.9 | 52.0 | 81.2 | 73.8 | 65.8 | 95.5 | 94.5 | 96.0 | 87.7 | 93.4 | 93.3 | 87.3 | 72.7 | 55.4 | 77.2 |
| MRN | 58.5 | 55.6 | 80.7 | 72.8 | 66.9 | 95.5 | 94.9 | 99.2 | 91.0 | 95.1 | 94.4 | 89.2 | 75.8 | 59.4 | 79.7 |
| bMTL | 56.6±0.3 | 51.8±0.5 | 80.9±0.3 | 72.9±0.4 | 65.6±0.1 | 95.0±0.2 | 94.5±0.7 | 96.0±1.2 | 86.8±0.2 | 93.1±0.1 | 92.9±0.5 | 86.5±0.2 | 71.9±0.8 | 56.0±0.8 | 76.8±0.3 |
| **VMTL-AC** | 62.3±0.1 | 52.4±0.3 | 82.0±0.2 | 75.9±0.4 | 68.2±0.1 | 95.2±0.3 | 95.7±0.5 | 99.4±0.3 | 90.7±0.4 | 95.2±0.2 | 92.9±0.3 | 87.7±0.2 | 77.0±0.7 | 60.1±0.7 | 79.4±0.2 |
| **VMTL** | 64.0±0.3 | 53.3±0.3 | 82.5±0.2 | 76.6±0.3 | **69.1±0.1** | 95.6±0.1 | 95.8±0.4 | 99.2±0.6 | 90.6±0.5 | **95.3±0.1** | 94.0±0.2 | 89.7±0.3 | 77.9±0.4 | 59.7±0.5 | **80.3±0.2** |

Table 16. Performance comparison of different methods on the large-scaled dataset *DomainNet* for multiple tasks: Clipart (C), Infograph (I), Painting (P), Quickdraw (Q), Real (R) and Sketch (S).

| Methods | 2% | | | | | | | 4% | | | | | | |
|---|---|---|---|---|---|---|---|---|---|---|---|---|---|---|
| | C | I | P | Q | R | S | Avg. | C | I | P | Q | R | S | Avg. |
| STL | 20.5 | 6.6 | 26.0 | 7.2 | 56.5 | 14.0 | 21.8 | 23.0 | 7.1 | 30.4 | 5.2 | 58.7 | 16.3 | 23.5 |
| VSTL | 33.8 | 12.3 | 37.1 | 23.7 | 65.3 | 23.2 | 32.6 | 26.4 | 8.9 | 30.9 | 20.2 | 60.7 | 17.8 | 27.5 |
| bMTL | 34±0.1 | 11.9±0.3 | 36.8±0.1 | 24.7±0.2 | 64.9±0.2 | 23.1±0.3 | 32.6±0.1 | 26.2±0.2 | 8.7±0.2 | 30.3±0.2 | 20.7±0.1 | 59.6±0.2 | 17.8±0.2 | 27.2±0.1 |
| **VMTL-AC** | 31.7±0.2 | 10.4±0.3 | 35.3±0.2 | 21.3±0.2 | 62.7±0.1 | 21.4±0.2 | 30.5±0.2 | 24.9±0.0 | 8.0±0.2 | 29.0±0.1 | 18.2±0.1 | 58.1±0.1 | 16.3±0.1 | 25.8±0.1 |
| **VMTL** | 35.7±0.1 | 14.3±0.2 | 40.0±0.2 | 18.5±0.1 | 65.4±0.2 | 25.5±0.1 | 33.2±0.1 | 30.9±0.1 | 11.9±0.0 | 35.4±0.2 | 15.7±0.1 | 61.8±0.1 | 21.8±0.2 | **29.6±0.1** |

