# OpenReview forum: "Variational Multi-Task Learning"
_ICLR.cc/2021/Conference — Reject_

### Official Review · AnonReviewer1 · 2020-10-27
**Very nice paper that looks at MTL from a variational inference perspective motivated for the small-data regime**

**Rating:** 8
**Confidence:** 5

**Review:**

[Summary]

In this paper, the authors propose to perform multi-task learning by casting as a variational inference problem. The key of the technique lies in how task relationships are modelled, through clever use of priors. In this paper, the priors for each task are conditioned on other tasks to enable knowledge transfer. The authors test their method in the small data-regime and show consistently better performance than the baselines.

[Decision]
I think this is a strong paper. The novelty is clear, the method well presented and the experimental section is good. Whilst grading an initial "7", I would potentially like to see (if possible) more results on bigger datasets, instead of restricting to 20%, to better understand the strength of the method when compared to other MTL baselines.

[Decision after reading rebuttal]
The authors appropriately addressed my concerns. The added experiments definitely reinforce the results of the paper and the added text help clarity. I recommend acceptance of this paper and would argue that its quality is reflect by an 8, up from my original score of 7.

[Main points]
1. The decision to cast multi-task learning as a variational inference problem where task relatedness is modelled through use of priors and posteriors is very interesting and well developed.
2. This paper is very well written, and whilst technical, is mostly easy to follow.
3. The ablation studies in addition to the experiments are extensive and sufficient to demonstrate: i) the advantages of the Bayesian modelling and ii) strengths of the method

[Questions]
1.  Why is a Gumbel-Softmax necessary to learn the mixing weights when updating the priors based on the other tasks? From what I can see, this is not a stochastic process that requires sampling from a Categorical. Why not learn $\mathcal{A}$ simply through backprop?
2. Modelling task relatedness is performed through the other task posteriors in Equation 9/10. In cases where they may only be task interference such that tasks should not be shared, how is this dealt with? Would $\mathcal{A}$ collapse to a uniform prior?
3. I am confused about the implementation (Section 4.2.). Is the VGGNet used as the backbone of the inference network for $z$ whilst classifiers on top of this are $w$? You state in Section 2.3. that an MLP is used for amortised inference. A more detailed explanation of the implementation is required to gain a better understanding of how everything is set up.
4. I am a bit confused about how the amortised inference is performed. For a given task $t$, there are $C$ specific labels. In Equation 12, do you mean that that weights for different labels are drawn from different distributions?
5. Whilst not the focus of the paper, how does the method perform when there is more data accessible? It would be interesting to gauge the competitiveness of the method then. In Tables 3 and 4, if we compare VMTL vs MRN, the difference in performance is not so significant. This might point that Bayesian MTL methods, with clever prior construction are the way to go. Results on larger datasets might help ascertain better performance of the method.

[Other comments]

1. One notable reference is missing from the Related Works section; one that first proposed to use the Gumbel-Softmax for learning task relatedness in MTL in [1].
2. I would like to see a more in depth discussion between your method and [2, 3, etc.]. As your method works comparably to MRN and can be classified as MTL papers with clever priors, I would like to understand more the differences + pros/cons of techniques.
3. Is there a way to inspect what was learned to be shared?
4. It is mentioned in the introduction that a Bayesian framework would allow you to deliver uncertainty estimates over the predictions. Were you able to analyse this?
5. Would your method be amenable to a continual learning scenario where the posterior over all prior tasks could be used for a new task without suffering catastrophic forgetting?


[1] Bragman et al. Stochastic Filter Groups - https://openaccess.thecvf.com/content_ICCV_2019/papers/Bragman_Stochastic_Filter_Groups_for_Multi-Task_CNNs_Learning_Specialist_and_Generalist_ICCV_2019_paper.pdf
[2] MRN paper - https://arxiv.org/abs/1506.02117
[3] MTL with Dirichlet Process Priors - https://www.jmlr.org/papers/volume8/xue07a/xue07a.pdf

---

> ### Author Response · Authors · 2020-11-23
> **Response to AnonReviewer1 (2/2)**
>
> [Other comments]
>
> 1. Thank you, we have included the reference for discussion in Section 3.
>
> 2. We have added the discussions about the pros and cons of the references and their differences from our method in Section 3. We attach the discussions here for your quick reference.
>
> * MRN[2] vs Our VMTL
>
> **Difference**: MRN [2] explores tensor normal distribution as priors of network parameters in different layers to learn multilinear task relationships, while our VMTL proposes a Gumbel-softmax prior dependent on variational posteriors of related tasks to learn the relatedness among task in the latent representation space and the parameter space. Compared with MRN, our VMTL jointly models the uncertainty of representations and classifiers in one unified Bayesian model, which can effectively learn the model without suffering from overconfidence.
>
> **Pros**: The advantage of MRN is to explicitly model the positive and negative relations across features and tasks.
>
> **Cons**: MRN depends on the deterministic classifier for each task, providing no mechanism for the uncertainty in the parameter space.
>
> * SMTL[3] vs Our VMTL
>
> **Difference**: The proposed SMTL models [3] proposes a nonparametric hierarchical Bayesian model to avoid the high complexity of model parameters and are implemented with a deterministic inference method. While our VMTL addresses multi-task learning in a probabilistic inference framework by casting it as a variational Bayesian inference problem.
>
> **Pros**: The advantage of [3] is to model the parameters of individual tasks without any assumption about the functional form of the prior distribution.
>
> **Cons**: When the training size is small, [3] weakly finds the similarity between tasks and most of the tasks learn by themselves, therefore their method could degenerate to single-task learning.
>
> 3. We can check the learned $\pi$ to inspect what is learned to be shared. In the setting of this work, different tasks share the same label space. In this sense, what is learned to be shared in classifiers is the semantic knowledge about object categories. About representations, samples from the same class but different tasks share similar representations.
>
> 4. The Bayesian framework naturally produces probabilistic distributions over predictions. More concretely, it gives the probabilities of a sample being classified, which intrinsically incorporates the uncertainty.
>
> 5. Yes, it is amenable to continual learning, and using the posterior of one previous task as the prior of the current task can reduce catastrophic forgetting. We have added the discussion in Section 3.

---

> ### Author Response · Authors · 2020-11-23
> **Response to AnonReviewer1 (1/2)**
>
> We thank *AnonReviewer1* for the positive comments *"this is a strong paper. The novelty is clear. the paper is well presented and the experimental section is good."*
>
> [Questions]
>
> 1. i) The Gumbel-Softmax technique provides an effective way to learn binary values $\mathcal{A}\_{ti}$ that indicates the relatedness between tasks. We have clarified it in Section 2.2. ii) Yes, we compared the Gumbel-softmax prior with the prior learned directly through backprop as shown in Table 2. Our Gumbel-Softmax prior outperforms alternatives.
>
> 2. Thank you for your comment. The fundamental assumption in multi-task learning is that tasks are related and there is always positive transfer among them. In the setting of this work, tasks share the same label space, and therefore the case with only task interference would hardly happen. In case of only task interference, the KL term will degenerate to an $\ell\_2$ regularization on the representation and classifiers. We have added the discussion in Section 2.2.
>
> 3. We apologise for the confusion about implementation. We use VGGnet to extract the feature representation $\textbf{x}$ for each sample and then we achieve its latent representation $\textbf{z}$ using amortized inference by MLPs (Kingma et al., 2013). Specifically, the amortization networks, i.e., MLPs, take the mean feature representations as input and return the mean and variation of $\textbf{w}$. We have added the detailed explanation in Section 4.2.
>
> 4. Yes, you are right. For a given task, we use the amortized inference to generate the classifier weight for each specific class by using the mean feature representations in this class. This means weights are drawn from different distributions. We have clarified it in Section 2.3.
>
> 5. i) The results with more data accessible (up to 50%) are indeed shown in Fig. 2. We provided detailed experimental results with more data in Table 7 of Appendix B.1. Our VMTL still performs better than other methods, though the performance advantage becomes less significant with more data. The results further demonstrate the effectiveness of our VMTL for multi-task learning, especially when training data is limited. ii) We have conducted additional experiments on the large-scale DomainNet (Peng et al., 2019), where more training data is accessible due to the number of categories is large. We add more details about this dataset in Section 4.1.  The results are reported in Table 6 and Table 16 in the updated manuscript. Again, our VMTL consistently achieves the best performance on this dataset, which demonstrates its effectiveness on a larger-scale dataset as well.
>
> [*Peng et al. Moment matching for multi-source domain adaptation. In ICCV 2019.*](https://openaccess.thecvf.com/content_ICCV_2019/papers/Peng_Moment_Matching_for_Multi-Source_Domain_Adaptation_ICCV_2019_paper.pdf)

---

### Official Review · AnonReviewer3 · 2020-10-28
**Improper Bayesian model formulation, experiments do not have confidence intervals**

**Rating:** 5
**Confidence:** 4

**Review:**

##########################################################################
### Summary
The paper proposes a Bayesian formulation of multi-task learning in the classification scenario.  Specifically, the problem is cast at a variational Bayesian inference problem. The inter-dependency of the tasks is enforced through Gumbbel-softmax priors, with the weights learned from the data. The Bayesian formulation leads to better performance when the amount of training data is low, as shown on three benchmark datasets for image classification.

##########################################################################
### Reasons for score
 Overall, I vote for reject. The methodology is not formulated in a proper Bayesian way as claimed.
 The experiments, while covering a wide range of datasets and scenarios, do not have any error bars
 and hence have very low statistical significance.

##########################################################################
### Pros
1) The use of the Bayesian approach is well motivated for the case of limited data.
2) The explicit inter-task dependency of the classifier weights is introduced, in addition to the
mode standard approach of interdependent features.
3) The related work section is quite thorough.
4) The experiments cover a wide range of datasets and analyse different aspects of the model: effect
of variational approximations on latent representations and the classifier weights, particular
choice of the prior, different amounts of training data available.

##########################################################################
### Cons
1) While other Bayesian multi-task models are mentioned on the related work, their applicability
for the task at hand is not addressed and the need for this particular approach is not motivated.
2) The model is not formulated in a standard Bayesian way. There is no clear separation between the model and the inference scheme. Specifically, the prior for one task is dependent on the variational posterior for the other tasks and is learned, hence it is not a proper prior.
3) The experiments do not have any error bars, only one value (i.e. one run). Thus the experiments
only weakly support the claims and are not statistically significant.

##########################################################################
### Some typos/ minor comments
Eq. (2): Use a different letter in the product, not t

Fig. 1: What do the dashed lines show? Is this an illustration of the model or the inference?

Sec. 4.2 Implementation: You say about using VGGnet as a  "feature extractor in your architecture".
The phrasing suggests you use it to extract z. I assume you pre-extract x. Clarify, please.

Supplement: Eq. (15): The very last term should have expectation over both q(w) and q(z|x)

Supplement B.3: "the prior of current takes is " -> "the prior of current tasks is"

---

> ### Author Response · Authors · 2020-11-23
> **Response to AnonReviewer3**
>
> We thank *AnonReviewer3* for stating our Bayesian approach is well motivated, our review of related work is thorough, and our experiments cover a wide range of datasets that analyse different aspects of the model.
>
> [Questions]
>
> 1. We have clarified in Section 1 and Section 3 that other Bayesian multi-task models, e.g., MTRL and MRN, are also applicable to the data setting and included in our comparison (Table 14). The need for our particular approach has two major motivations: (i) The motivation of our VMTL is to leverage Bayesian modeling to handle the great challenges caused by limited data in multi-task learning. (ii) The motivation of the introduced variational inference is that it allows us to specify the priors by depending on variational posteriors of related tasks; in doing so, we are able to explore and leverage task relatedness among classifiers and feature extractors in a general principled way.
>
> 2. We apologise for causing any confusion or misunderstanding here. We further clarify it here for you and update our manuscript accordingly in Section 2.1. Our Bayesian formalism for multi-task learning would look a bit different from conventional Bayesian formalism for single-task learning, in which the prior is usually taken as a non-informative normal Gaussian distribution. However, under the multi-task learning setting, since data $\mathcal{D}\_{1:T}$ from all tasks are observed, for each task $t$, the posterior over the parameter of the task is $p(\mathbf{w}\_t|\mathcal{D}\_{1:T})$ and by applying Bayes' rule, we obtain the prior $p(\mathbf{w}\_t|\mathcal{D}\_{1:T}\backslash {\mathcal{D}\_t})$ for task $t$. The prior depends on information from the other related tasks in the form of posteriors to leverage shared knowledge. Actually, priors serve as regularisation in Bayesian inference and provide a principled way of sharing information across multiple tasks. Note that in the optimization, multiple tasks are optimized one by one in each iteration. We have added this discussion in Section 2.2.
>
> 3. We have added the requested error bars with the 95% confidence level in all tables of the updated manuscript.  Also with the error bars our proposal is still consistently better than alternatives. Thank you.
>
> [minor comments]
>
> 1. Thank you for pointing this out. It has been fixed.
>
> 2. The two dashed lines show the prior of the current task depends on the posteriors of other tasks for classifiers and representations. Fig. 1 is an illustration of our model. We have clarified this in the corresponding caption.
>
> 3. Indeed, we use VGGnet to pre-extract $\mathbf{x}$, from which we infer $\mathbf{z}$. We have clarified it in Section 4.2 of the updated manuscript.
>
> 4. Thank you for your careful reviews. We have fixed the typos.

---

### Official Review · AnonReviewer4 · 2020-10-28
**Novel approach for multi-task learning using Bayesian inference framework - Good experimental results**

**Rating:** 7
**Confidence:** 3

**Review:**

This paper presents a variational based approach for multi-task learning for the setting where there is a limited amount of training data and for each related task. Prior distributions on model parameters are defined using weighted approximate posterior distributions of other related tasks. This has been done both for classification parameter (w) and latent features (z). The mixing weights are learned in the main optimization using Gumble-softmax prior for each task. The classifier parameters are learned with amortized inference to make posterior inference more effective. The ELBO is learned using MCMC samples using samples partially taken using the reparametrization trick. The experiment section compares the performance of the proposed method against previous multi-task learning approaches.

The idea of using a variational approach for multi-task learning is novel and interesting. Using posterior distributions of related tasks in a weighted sum for defining prior distributions for parameters of each task is an interesting method that could also inspire more future work in this area.  In terms of the learning algorithm, there are no significant contributions related to novelty, but incorporating various details such as a Gumble-softmax weighting approach, amortized learning, and reparametrization trick have helped this proposed approach to form an overall good performance.

Proposing this model for multi-task learning problems in which there are limited input samples for each of the related tasks is a problem that has not been explored enough before in the literature despite its importance for many domain applications. Focusing on this problem is an aspect that makes this paper have a high impact on the field.

The related work section gives a good overview of the previous and current approaches for multi-task learning from classical penalty driven models, to Bayesian approaches and deep learning models.

The experimental results show a clear improvement of VMTL over the current baselines. I liked that the authors have analyzed the importance of each of the pieces that are used in the model or in the learning algorithm such as the effectiveness of the Bayesian approximation and the choice of softmax for priors.

The paper is clearly written and organized. There are a couple of typos (takes instead of task in section 2.1 and B.3),..

Aside from a brief mentioning of cost reduction due to amortized learning, I don't see any computational time analyst. I think it would be very helpful if the authors could provide more information regarding running times compared to the baselines.

Also, MCMC methods tend to result in high variances especially in the limited data domain. Seeing some analysis on how the variance is affected by the hyperparameter choices and the temperature parameter would be very informative.

I would be interested to know the results for other multi-task learning-related tasks such as representation learning or robustness against adversarial attacks as the scope of this work doesn't have to be limited to the classification accuracies. Other applications such as NLP related ones could be interesting too as many multi-task learning approaches have been successfully applied there, and it would be nice to see how this could compare to those.

Overall, in my opinion, this paper presents a novel approach for a significant problem, and the experimental results support the claims. That is why I accept this paper.

---

> ### Author Response · Authors · 2020-11-23
> **Response to AnonReviewer4**
>
> We thank *AnonReviewer4* for stating  *"the idea of using a variational approach for multi-task learning is novel and interesting".*
>
> **Q**: *"``Aside from a brief mentioning of cost reduction due to amortized learning, I don't see any computational time analyst. I think it would be very helpful if the authors could provide more information regarding running times compared to the baselines."*
>
> **A**: We have added the suggested experiment for computational time analysis as shown in Fig. 3 of the updated manuscript. The amortized learning converges faster than non-amortized learning, which shows the computation benefit by amortized learning. This discussion is added in Section 4.3. Thank you.
>
> **Q**: *"MCMC methods tend to result in high variances especially in the limited data domain. Seeing some analysis on how the variance is affected by the hyperparameter choices and the temperature parameter would be very informative."*
>
> **A**: Indeed, the better performance of stochastic gradient ascent crucially depends on the lower variance of the gradients. In MC sampling, to reduce the gradient variance, we adopt the local reparameterization trick (Kingma et al., 2015). The number of MC samples is the hyperparameter that can affect the variance and set to be 10 for computational efficiency. In Gumbel-softmax priors, when the temperature is small, the variance of the gradients can be large. In practice, we start with a high temperature and gradually anneal it to a small but non-zero value. The Gumbel-softmax technique offers a derivative gradient estimator with a low-variance path for the categorical distribution. We have added this analysis in Section 2.4 and Section 4.2. Thank you.
>
> **Q**: *"I would be interested to know the results for other multi-task learning-related tasks such as representation learning or robustness against adversarial attacks as the scope of this work doesn't have to be limited to the classification accuracies. Other applications such as NLP related ones could be interesting too as many multi-task learning approaches have been successfully applied there, and it would be nice to see how this could compare to those."*
>
> **A**: Thank you for these great suggestions. Following your suggestions, we try our method in the scenario of adversarial attacks.
> We conduct experiments to show the robustness of our methods against adversarial attacks. In our experiments, the adversarial attack is implemented by the fast gradient sign method (Goodfellow et al., 2015) where $\epsilon$ denotes the noise level. We evaluate our proposed VMTL, VMTL-AC, and the basic multi-task learning(bMTL) on the Office-home dataset. As shown in Fig. 4 of Appendix B.5, under different noise levels, VMTL outperforms bMTL. As the noise level increases, the variant of our method VMTL-AC is more robust and significantly outperforms the baseline multi-task learning model. The possible reason could be that VMTL-AC leverages the center feature (training samples) to generate the classifier which makes up for the lost information induced by the gradient noise. We have added the discussion in Section 4.3 and experimental details in the Appendix B.5.
>
> In addition, though the major focus of the work is on the multi-task classification problems, our VMTL could also be used to solve other interesting applications, e.g., NLP problems. We would like to explore this in  future work.
>
> [*Goodfellow et al., Explaining and harnessing adversarial examples, In ICLR 2015.*] (https://arxiv.org/pdf/1412.6572.pdf)
>
> **Q**: *"There are a couple of typos (takes instead of task in section 2.1 and B.3)"*
>
> **A**: We have fixed the typos. Thank you.

---

### Official Review · AnonReviewer2 · 2020-10-30
**A variational framework for multi-task learning**

**Rating:** 7
**Confidence:** 3

**Review:**

Summary:

This paper presents a variational framework for multi-task learning, where for each image classification task, the feature representation z and the classification parameter w of the other tasks serve as the prior for the counterparts of the task. The priors are engaged as the regulariser in the variational inference framework, making the features and the classification parameters of the tasks close to each other. The proposed method outperforms several baselines on several image classification benchmark datasets.

Pros:

- The proposed way of using a Bayesian (variational) approach where the feature and classification parameters of other tasks are leveraged as priors is quite intuitive. As priors naturally serve as regularisations in Bayesian inference, it is natural and might be a principled way of sharing information across multiple tasks. The formulation of multi-task learning with the Bayesian framework might not be completely new in the conventional Bayesian settings, but it seems to be new in the auto-encoding/amortized variational inference settings. Although I'm not an expert in the domain of multi-task learning, I didn't find a similar point of view in the literature.

- I also think the proposal of the prior construction in Eq (9) and Eq (11) is interesting and clever, which is conducted by weighting the posteriors of the other tasks.

- It is also a plus that the authors have done comprehensive oblation study of the proposed framework, showing the different contributions of the components.  To me, the components form an elegant model.

- The paper is easy to follow.

Cons:

- The use of the Gumbel softmax in Eq (9) and Eq (10) is confusing to me. Motivation, intuition, and the learning of the A_ti are a bit unclear. Which parameter is learned in the end? Is it \pi_ti? If so, the \pi_ti is a kind of free parameter to learn. Then why is \pi_ti put through a Gumbel softmax?

- I also feel that the datasets used in the paper are a bit small-scaled in today's machine learning/deep learning areas.

---------------------------------------------------------------------------------------------------------------------------

The authors have addressed my previous concerns. The experiments on a larger dataset are a plus. Therefore, I would recommend an acceptance of the paper.

---

> ### Author Response · Authors · 2020-11-23
> **Response to AnonReviewer2**
>
> We thank *AnonReviewer2* for stating our proposal is *"interesting and clever"* and *"the components form an elegant model"*.
>
> 1. The motivation of learning Gumbel-softmax priors is to effectively explore task relatedness.
> Intuitively, $\mathcal{A}\_{ti}$ is a binary value that indicates whether two tasks are correlated or not. To enable learning this binary value with back-propagation, we introduce the Gumbel-softmax technique with the learnable parameters $\pi\_{ti}$, which generates $\mathcal{A}\_{ti}$ by sampling as shown in Eq. (10), where $\pi\_{ti}$ denotes the probability of two tasks are correlated.  We have clarified this in Section 2.2.
> Yes, $\pi\_{ti}$ can be learned directly by treating it as a kind of free parameter. Indeed, our experimental results in Table 2 demonstrate that learning with Gumbel-softmax outperforms directly learning $\pi\_{ti}$ (Learnable weighted).
>
> 2. We have conducted additional experiments on the large-scale DomainNet (Peng et al., 2019), which contains approximately 0.6 million images with 345 categories from six distinct domains. We add more details about this dataset in Section 4.1. To the best of our knowledge, this dataset has not been used for multi-task learning. The results are reported in Table 6 and Table 16 in the updated manuscript. Again, our VMTL consistently achieves the best performance on this dataset, which demonstrates its effectiveness on a larger-scale dataset as well.
>
> [*Peng et al. Moment matching for multi-source domain adaptation. In ICCV 2019.*](https://openaccess.thecvf.com/content_ICCV_2019/papers/Peng_Moment_Matching_for_Multi-Source_Domain_Adaptation_ICCV_2019_paper.pdf)

---

### Decision · Program_Chairs · 2021-01-07
**Final Decision**

**Decision:**

Reject

**Comment:**

This paper presents a probabilistic model for multitask learning with representation learning. The basic idea is to share information across tasks by making the prior over the model parameters of one task conditioned on a convex combination of the variational posteriors of the other tasks.

While some of the reviewers gave high scores and recommended acceptance, one of the reviewers (AnonReviewer3) had some pertinent concerns which lingered even after author response. In particular, AnonReviewer3 mentions that since the prior of one task is conditioned of the variational posteriors of the other tasks, the method is not a proper Bayesian approach. I also read the paper and agree with the assessment. Indeed, the common Bayesian way for multitask learning is to couple the tasks purely based on a prior that encouraging sharing across tasks instead of having task-specific prior that depend on the variational posterior of other tasks as is being done in this paper.

I also read the reviews and the author response and have some other concerns as well:

- There is a huge amount of prior work on multitask learning, both non-Bayesian as well as Bayesian. Although the paper cites several of those it is disappointing that none of the baselines are Bayesian. Even the non-Bayesian baselines aren't the state-of-the-art recent methods, which is disappointing given the extensive body of prior work in this area.

- The rebuttal wrongly claims MTRL and MRN to be Bayesian methods (included in Table 14 as baselines) whereas they only have a probabilistic formulation and only do point estimation. At a minimum, the paper should show comparison with some Bayesian multitask learning approaches (e.g., shared hierarchical priors, or task clustering, etc). The baselines such as MTRL and MRN aren't among the strongest ones out there.

- The paper's title is way too generic. There are several multitask learning papers that use variational inference for a Bayesian model. Moreover, given that the basic formulation itself is a bit problematic to be called Bayesian, the title in some sense is also misleading.

Due to the above issues, I don't think the paper can be accepted in its current form.